# Anomalous Left Coronary Artery from the Pulmonary Artery: How to Diagnose and Treat

**DOI:** 10.3390/jpm13111561

**Published:** 2023-10-31

**Authors:** Elaina A. Blickenstaff, Sean D. Smith, Frank Cetta, Heidi M. Connolly, David S. Majdalany

**Affiliations:** 1School of Science, Marshall University, Huntington, WV 25755, USA; 2Division of Cardiology, Mayo Clinic, Scottsdale, AZ 85259, USA; 3Division of Cardiology, Mayo Clinic, Rochester, MN 55905, USA

**Keywords:** anomalous coronary artery, adult congenital heart disease, anomalous left coronary artery from the pulmonary artery (ALCAPA)

## Abstract

Anomalous origin of the left coronary artery from the pulmonary artery (ALCAPA) is a rare coronary anomaly that can present in childhood or adulthood with a multitude of symptoms depending on the age of presentation. It should be suspected in infants presenting with heart failure in the setting of left ventricular systolic dysfunction and associated mitral regurgitation from papillary muscle ischemia. Adults with ALCAPA may present with cardiac ischemic symptoms. Prompt diagnosis with echocardiography and cross-sectional chest imaging is important to guide surgical intervention and improve the patients’ survival and prognosis. The goal of surgery is to establish a dual-coronary system with mid-term results revealing progressive recovery of left ventricular function and improvement in mitral regurgitation. Patients with ALCAPA should maintain life-long follow-up with a cardiologist with congenital heart disease expertise for surveillance of post-operative complications.

## 1. Introduction

Anomalous origin of the left coronary artery from the pulmonary artery (ALCAPA) syndrome is a rare congenital heart condition that affects approximately 1 in 300,000 live births [1]. It comprises between 0.24 and 0.46% of all congenital heart disease [2]. First clinically described in 1933 by Edward Bland, Paul Dudley White, and Joseph Garland, ALCAPA is, therefore, also known as Bland–White–Garland syndrome; it is most often an isolated congenital heart defect but can present with other associated congenital heart diseases including atrial septal defect, ventricular septal defect, patent ductus arteriosus, tetralogy of Fallot, aorto-pulmonary window, and coarctation of the aorta. Embryologically, the cells of the capillary plexus surrounding the pulmonary artery and aorta fail to reach and/or penetrate the two normal coronary origins in the setting of deficiency of the vascular endothelial growth factor-C, leading to anomalous coronary origin from the pulmonary artery. ALCAPA may result from the involution of the aortic buds, which eventually form the coronary arteries, with concomitant persistence of the pulmonary buds as well as abnormal separation of the conotruncus from the aorta and pulmonary artery [3,4].

Individuals with ALCAPA most often present in infancy after pulmonary vascular resistance drops, resulting in retrograde flow from the left coronary artery to the pulmonary artery (left to right shunt); this, in turn, leads to reduced coronary perfusion, myocardial ischemia, and resultant ventricular dysfunction as well as other complications including dysrhythmias, mitral regurgitation, and sudden cardiac death [5,6]. However, adequate collateral circulatory development can provide adequate coronary perfusion, resulting in a relatively asymptomatic clinical course in a select few, allowing for a late onset presentation, including during adulthood. Right coronary artery dominance, minimal coronary artery steal from the pulmonary artery due to ostial left main stenosis or restrictive opening in the pulmonary artery, and development of a systemic blood supply to the left coronary artery from bronchial artery collaterals are additional factors that enable patients with ALCAPA to survive into adulthood. Symptoms vary with age and the extent of collateral development; historically, the patient may exhibit symptoms of angina pectoris, dyspnea, syncope, and palpitations. Physical examination is generally unremarkable but may reveal a heart murmur consistent with mitral regurgitation, abnormal pulmonary examination with crackles consistent with pulmonary edema, and signs of heart failure (e.g., elevated jugular venous pressure, poor perfusion, peripheral edema.). The recommended treatment for ALCAPA is surgery that reestablishes a dual coronary system. Without surgical intervention, those with ALCAPA are at risk of cardiac death, myocardial infarction, severe mitral regurgitation, decompensated heart failure, and ventricular arrhythmias from scarred tissue.

## 2. ALCAPA Subtypes

ALCAPA is classified into two subtypes by age: those who present in infancy (85%) and, more uncommonly, those who present as older children, teens, or adults (15%), i.e., the so-called adult subtype [4,7].

Infant ALCAPA typically manifests about 2–3 months after birth and is characterized by presenting symptoms of congestive heart failure as a result of myocardial ischemia in the setting of little to no coronary collaterals. Common signs of infant ALCAPA are dyspnea, tachypnea, prolonged or difficult feeding time, pallor, diaphoresis, delayed development, and failure to thrive [7]. Evaluation of the patient may reveal varying degrees of mitral regurgitation and left ventricular dysfunction. Mitral regurgitation may be functional, secondary to dilated left ventricle and annulus, or a result of papillary muscle ischemia and fibrosis [4,8]. Pathophysiologically, this occurs as a result of changes in the pulmonary artery pressure after birth; in utero, pulmonary and aortic pressures are essentially equivalent, and physiologic shunting across a normal patent ductus arteriosus and patent foramen ovale allows for normal coronary perfusion. However, postpartum, the pulmonary artery pressure drops drastically, and physiologic shunts typically close. In this setting, aortic pressure far exceeds pulmonary artery pressure, which prompts preferential blood flow from the left coronary artery in the lower-pressure pulmonary circulation rather than the high-resistance myocardial circulation. As a result, downstream perfusion of the left coronary system is poor and results in myocardial ischemia. The degree of collateral development determines the age of presentation in this case: more extensive collateralization from the right coronary artery allows for this lesion to remain undiagnosed for a period of time, whereas inadequate collateral blood flow results in early presentation and diagnosis [7].

Timely surgical correction of infant ALCAPA often addresses the sequelae listed above and can result in complete recovery of left ventricular function. Mitral valve repair is usually not needed in infancy. Without surgical correction, infant mortality is reported to be as high as 90% in the first year of life [4,7].

Adult ALCAPA is characterized by extensive and dilated coronary collateral arteries, cardiomegaly, mitral regurgitation, and/or reduced left ventricular ejection fraction [7]. Adult ALCAPA may be discovered at any point in adulthood and is often an incidental finding when symptoms such as angina pectoris prompt an investigation. Yau et al. reviewed 151 adult cases of ALCAPA and determined the average age of diagnosis to be 41 years, with 48 patients being older than 50 years and a greater than 2:1 female-to-male ratio [8]. However, in 2015, Sinha et al. described the diagnosis of ALCAPA in a 75-year-old woman who presented with progressive exertional angina over several months [2]. More active individuals may show symptoms earlier in adulthood, but the time of diagnosis varies greatly. Common symptoms of adult ALCAPA, when present, include palpitations from dysrhythmias, syncope, dyspnea, chest pain, and fatigue. ALCAPA is a well-known and important cause of sudden cardiac death in adult patients and should be considered in the differential diagnosis of cardiac arrest. The risk of sudden cardiac death tended to decrease with age, with autopsy studies of adults with unrepaired ALCAPA revealing the average age of death to be 35 years [4]. Surgical correction of adult ALCAPA can often address the issues listed above but may not result in complete recovery of left ventricular function due to chronic myocardial ischemia before evaluation and treatment. 

## 3. Diagnostic Testing

Invasive coronary angiography (ICA) was the standard for ALCAPA diagnosis as it depicted the course of the anomalous coronary artery; however, it has been largely replaced by noninvasive diagnostic testing. However, during the course of a workup for angina, the patient may undergo ICA and be found to have ALCAPA incidentally. Typical findings on ICA include a dilated and tortuous right coronary artery with multiple collaterals to the left coronary system; anomalous flow into the pulmonary artery can also be seen (Figure 1) [7]. Transthoracic echocardiogram (TTE) with color Doppler is a safe, readily available, inexpensive, and portable noninvasive method for initial investigation used in all patients (Table 1). Echocardiographic findings indicative of ALCAPA include visualization of the left coronary artery originating from the pulmonary artery (Figure 2), retrograde flow from the left coronary artery to the pulmonary artery (Figure 3), dilated and tortuous right coronary artery, lack of the left coronary artery at aortic origin, significant and dilated collateral coronary arteries, mitral regurgitation, left ventricular dysfunction with regional wall motion abnormalities, and enhanced echogenicity of papillary muscles [9,10,11]. The parasternal short-axis acoustic window on TTE provides the best views of the origins of the coronary arteries. Increased flow in the minor coronary arteries due to the collateral flow from the right to the left coronary artery may be detected by lowering the Nyquist limit. In younger patients with dilated right coronary artery, collateral arterial flow in the ventricular septum may be seen and misdiagnosed as multiple trabecular ventricular septal defects. In order to distinguish between these two, pulse-wave Doppler should be performed: continuous flow in the collateral vessels is noted in patients with ALCAPA, while systolic flow into the right ventricle is noted in patients with ventricular septal defects [11]. Transthoracic echocardiogram, while fast and easy, has poor spatial resolution, making it difficult to identify and visualize the arteries; thus, it should be supplemented with further diagnostic testing such as computed tomography angiography (CTA), magnetic resonance angiography (MRA), or ICA [4].

Electrocardiographic findings suggestive of ALCAPA are abnormal deep or wide Q waves, inverted T waves, and poor R wave progression in leads I, aVL, and precordial leads V4 to V6 (Figure 4) [14]. Additionally, left ventricular hypertrophy and myocardial injury patterns may be present. Noninvasive ischemic evaluation, such as cardiopulmonary exercise testing, stress echocardiography, and myocardial stress perfusion, would reveal ischemia. In fact, previous studies had revealed that stress ECG and stress imaging tests were positive for ischemia in 85–87% of patients with ALCAPA [4,8].

Computed tomographic angiography provides superior visualization of the coronary arteries and can be used for the definitive diagnosis of ALCAPA (Table 1), anatomic assessment, and postoperative follow-up. When CTA is utilized (Figure 5), direct visualization of the left coronary artery originating from the pulmonary artery, a dilated right coronary artery with extensive coronary collateral arteries, abnormal left ventricular wall movement, and dilated bronchial arteries—which act as systemic collaterals—denote ALCAPA [7]. In ALCAPA, the left coronary artery arises from the left inferolateral aspect of the main pulmonary artery just after the pulmonary valve and courses toward the interventricular groove before branching into the left anterior descending artery and the left circumflex artery. In infants, the right and left coronary arteries may appear normal. However, the coronary arteries are dilated and tortuous in adults, with dilated intercoronary collateral arteries coursing along the epicardial surface of the heart or within the interventricular septum [7].

Magnetic resonance angiography provides the best functional visualization of the coronary arteries. It is superior to CTA for pediatric patients, as there is no exposure to ionizing radiation and no requirement for low heart rate (Table 1). In MRA, left ventricular hypertrophy secondary to chronic myocardial hypoperfusion, mitral insufficiency or prolapse, myocardial ischemia, left ventricular wall motion abnormalities, and delayed subendocardial enhancement imply ALCAPA [7,15]. Delayed subendocardial enhancement is due to subendocardial infarction and may be predictive of the onset of malignant arrhythmias, prompting consideration of surgical repair. The retrograde flow from the left coronary artery to the main pulmonary artery, which represents the coronary steal phenomenon, can be depicted on MRA. Moreover, the direction and volume of this flow can be quantified on fast cine phase-contrast images [7].

**Figure 4 jpm-13-01561-f004:**
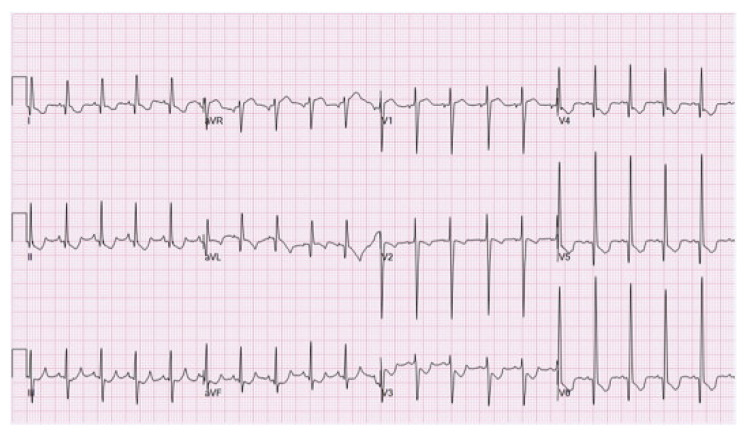
Electrocardiogram of anomalous left coronary artery from the pulmonary artery reveals Q waves and ST depressions and T wave changes in the anterolateral leads (leads I, AVL, V5, and V6). Open access figure from Al-Fayyadh, M.; Alwadai, A.; Al Huzaimi, A.; Al Halees, Z. Near missed reversible cardiomyopathy: The value of the electrocardiogram. *Int. J. Pediatr. Adolesc. Med.*
**2015**, *2*, 29–33. [16].

**Figure 5 jpm-13-01561-f005:**
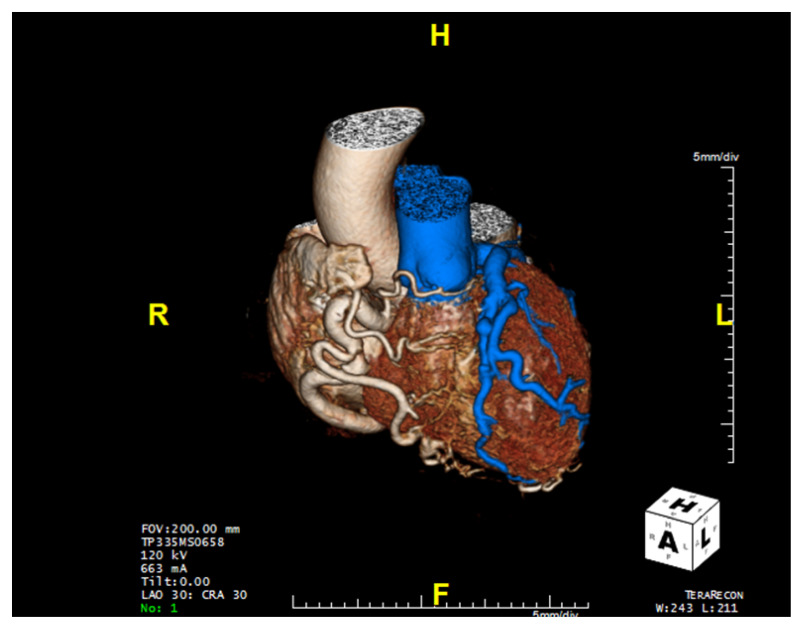
Computed tomography of the heart revealing anomalous left coronary artery from the pulmonary artery (blue) as well as the dilated and tortuous right coronary artery.

## 4. Treatment

The 2018 Adult Congenital Heart Disease Guidelines from the American Heart Association and American College of Cardiology recommend surgery for ALCAPA patients regardless of age and symptoms due to lifelong risk of ischemia, ventricular dysrhythmias, and sudden cardiac death [4,8,17]. Surgical approaches can be divided into either a one-coronary system repair or a two-coronary system repair. Simple ligation of the anomalous artery, a type of single-coronary repair, was common in the past. Though no longer performed exclusively, simple ligation can be used to delay proper surgical correction and prevent coronary steal in severely ill infants with ALCAPA. Delaying the more intensive corrective surgery could be beneficial in some circumstances because it allows critically ill infants to recover and further develop, lessening the risks of the operation and facilitating recovery [18].

The modern objective of all ALCAPA surgeries is to reestablish a dual coronary system that provides oxygenated blood to the entire heart. Various surgical approaches for ALCAPA have been utilized, including direct reimplantation of the coronary artery into the aorta (coronary button transfer), transpulmonary baffling (Takeuchi procedure), subclavian-left coronary anastomosis, and coronary artery bypass grafting using a saphenous vein graft or left internal mammary artery with ligation of the anomalous left coronary artery [7,14,19,20]. Cardiac transplantation is reserved for patients with severe left ventricular dysfunction and refractory heart failure. 

Coronary button transfer is the preferred method in infants, when feasible, as this provides near-physiologic circulation [7,14]. This method involves transecting the main pulmonary artery, harvesting a circular portion of the pulmonary artery surrounding the anomalous left coronary artery ostium (the so-called “button”), creating an opening in the aorta, anastomosing the left coronary artery into the aorta in an end-to-side fashion, and reconstructing the main pulmonary artery. Utilizing this method has excellent long-term results with low rates of ostial stenosis (Figure 6) [20].

Alternatively, when the length of the left coronary artery is insufficient for a tension-free connection, a modification of direct reimplantation called the two-flap technique can be employed. The two-flap technique requires harvesting the anomalous left coronary artery with a pulmonary artery flap. Subsequently, an aortic wall flap is created, and the two flaps are sutured together to lengthen the left coronary artery, producing tension-free anastomosis (Figure 7) [21].

The Takeuchi technique requires the creation of an intrapulmonary baffle to connect the anomalous left coronary artery and aorta. An aortopulmonary window is made first; then, the baffle is constructed by suturing an anterior pulmonary flap to the posterior wall of the pulmonary artery. The pulmonary artery is then repaired with a pericardial patch (Figure 8) [15].

In older patients, direct reimplantation is not always an option due to immobilization or friability of the left coronary artery. The Takeuchi technique or coronary artery bypass grafting is performed in such cases. Coronary artery bypass grafting involves ligation of the anomalous left coronary artery at its origin and connection of the left anterior descending artery and left circumflex artery to the aorta via an arterial or vein graft [7].

Corrective surgery is the standard for ALCAPA treatment in both infants and adults, but medications can be utilized in tandem with surgical intervention to ensure the best possible outcome. In a rare case reported by Sinha et al., an octogenarian woman was diagnosed with ALCAPA, and medication was used to maintain her stable condition rather than surgery because the risks of surgical intervention outweighed the potential benefits [2]. Additionally, Hu et al. suggest that infants suffering from acute myocardial ischemia caused by a lack of collateral arteries undergo a short preoperative period of drug treatment to recover cardiac function [5].

## 5. Long-Term Outcomes

Corrected ALCAPA results in generally positive long-term outcomes with low mortality rates. Hu et al. reported an 83.4% Kaplan–Meier survival rate one year post-operation in a study of 80 ALCAPA pediatric patients who underwent various corrective procedures. The median patient age was 7.8 months, and most patients underwent reimplantation of the coronary artery. There were 11 (13.8%) hospital deaths and 2 (1.3%) late deaths at 1-year follow-up in their series [5]. Mishra reported the outcomes of 105 patients with anomalous origin of the coronary artery from the pulmonary artery and 98 patients with ALCAPA; the median age at operation was 5.8 months with a median follow-up of 5.9 years. All patients underwent coronary reimplantation. In-hospital mortality was reported to be 8.5%, with no reported late deaths [19]. In most cases, the ejection fraction increased, left ventricle function recovered, mitral regurgitation improved, and symptoms ameliorated regardless of surgical method [23]. In the Mishra series, 33.6% of patients who underwent surgery during the newborn period (<6 months of age) had good recovery of left ventricular function (mean left ventricular ejection fraction from 50 to 55%) and had improvement in mitral regurgitation from moderate severity to mild residual regurgitation. In addition, 52% of the patients who underwent operations late in infancy (after 6 months of age) had residual impairment of left ventricular function (mean left ventricular ejection fraction from 40 to 50%) and mild-to-moderate mitral regurgitation; some experienced gradual improvement in their left ventricular function with subsequent follow-up, and three patients (3.4%) required mitral valve replacement after 3 years for unrepaired progressive mitral valve regurgitation [19]. However, the patient outcomes highly depend on the extent of myocardial damage prior to operation [15]. In a multicenter review study of 98 infant ALCAPA cases, Radman et al. found that approximately three years after corrective surgery, left ventricular function returned to normal in 98% of cases, whereas the trajectory of mitral regurgitation was more difficult to predict. The uncertainty regarding the improvement of mitral regurgitation was thought to be caused by potentially irreversible preoperative ischemia or suboptimal coronary perfusion after surgical correction due to loss of patency in the left coronary artery [24]. While the type of surgical correction has not been found to impact mortality, some methods are prone to greater complications.

Ligation of the ALCAPA, although a currently avoided technique, can be complicated by atherosclerosis, recanalization of the ALCAPA with resultant persistent shunt and coronary steal phenomenon, severe mitral regurgitation, and persistent silent ischemia, which may be a nidus for arrhythmias and sudden death [7].

Direct reimplantation and its modifications have few complications in infants, but in adults, left coronary artery tearing and bleeding can occur due to the friability of the left coronary artery and diminished elasticity when the anomalous coronary is mobilized for repair [7]. The Takeuchi technique can lead to supravalvular pulmonary stenosis due to pulmonary artery manipulation, tissue resection, baffle leaks causing coronary–pulmonary artery fistula, baffle obstruction, and aortic regurgitation [25]. In the Ginde et al. series of nine patients who underwent Takeuchi repair for ALCAPA, all eight surviving patients had some degree of main pulmonary artery stenosis, with moderate stenosis in two patients and severe stenosis in one patient. Moreover, three late survivors had a baffle leak [26]. Pulmonary artery stenosis can lead to the need for additional future interventions, including surgery or transcatheter procedures. Coronary artery bypass grafting, which is considered the preferred method in adults, is also generally well-tolerated but has been associated with graft occlusion and stenosis [7].

After surgical recovery, patients should have regular follow-ups to monitor cardiac function and complications. Because of their excellent spatial resolution, ECG-gated multidetector CTA and cardiac magnetic resonance imaging are valuable follow-up tools in adults after surgery [7]. Some patients may experience residual symptoms after surgery due to considerable myocardial damage or ischemia. An implantable cardioverter defibrillator may be implanted in select patients with severe left ventricular dysfunction to correct malignant dysrhythmias and prevent sudden cardiac death. However, the use of implantable cardioverter defibrillators (ICD) remains controversial, as it may lead to further complications such as infection. In a study of two patients with the adult form of ALCAPA conducted by Kubota et al., ICDs were implanted in both cases with varying results. Patient one had no complications and no ICD activations after 11.5 years. Conversely, patient two developed a methicillin-resistant Staphylococcus aureus infection on the ICD leads, tricuspid ring, and ICD generator, prompting the urgent removal of the device. It is imperative to note that patient two underwent reoperation and had other complications that increased the likelihood of an infection. Regardless of case details, the risk of infection associated with an ICD should always be considered prior to implantation [9].

Varying degrees of mitral regurgitation are frequently seen in ALCAPA patients, but simultaneous mitral valve valvuloplasty or annuloplasty remains controversial. Mitral regurgitation often improves after ALCAPA correction; thus, many recommend no mitral valve intervention [5,27]. However, if mitral regurgitation is severe, intervention may be warranted.

## 6. Conclusions

ALCAPA is a rare congenital coronary anomaly that results from abnormal left coronary artery origination from the pulmonary artery. This anomalous origin leads to coronary vascular bed ischemia in the left coronary artery territory and can lead to left ventricular dysfunction, mitral regurgitation, and sudden cardiac death. ALCAPA is subdivided into infant and adult subtypes, which can be distinguished by toleration of the coronary ischemia depending on the degree of collateral blood flow. Transthoracic echocardiography combined with other noninvasive tests such as CTA or MRA can be used to definitively diagnose the condition. Treatment of ALCAPA is surgical and aims to establish a two-coronary vascular bed. These techniques include button coronary transfer, transpulmonary baffle creation or flaps, and coronary artery bypass grafting with ostial ligation. While some patients may still experience symptoms after the operation, the majority of patients recover left ventricular function and are no longer at risk of sudden death. Patients should be followed postoperatively and monitored using serial cardiac imaging to detect various complications depending on the type of surgical correction. 

## Figures and Tables

**Figure 1 jpm-13-01561-f001:**
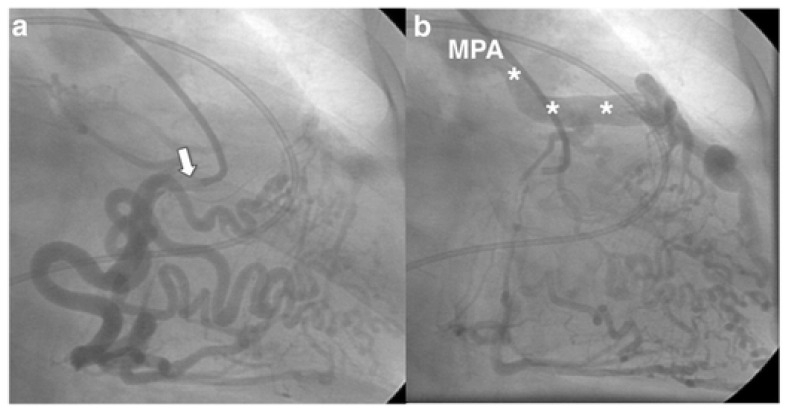
(**a**) Selective angiography of the right coronary artery (white arrow) revealing a dilated and tortuous vessel with multiple collaterals to the left coronary artery. (**b**) The injected intrave-nous contrast spills from the left coronary artery into the main pulmonary artery (MPA; asterisks), confirming anomalous left coronary artery from the pulmonary artery. Used with permission from Crean, A.; Ahmed, F.; Motwani, M. The Role of Radionuclide Imaging in Congenital Heart Disease. *Curr. Cardiovasc. Imaging Rep.*
**2017**, *10*, 38. https://doi.org/10.1007/s12410-017-9434-0; Executive Summary. *J. Am. Coll. Cardiol.*
**2019**, *73*, 1494–1563. [12].

**Figure 2 jpm-13-01561-f002:**
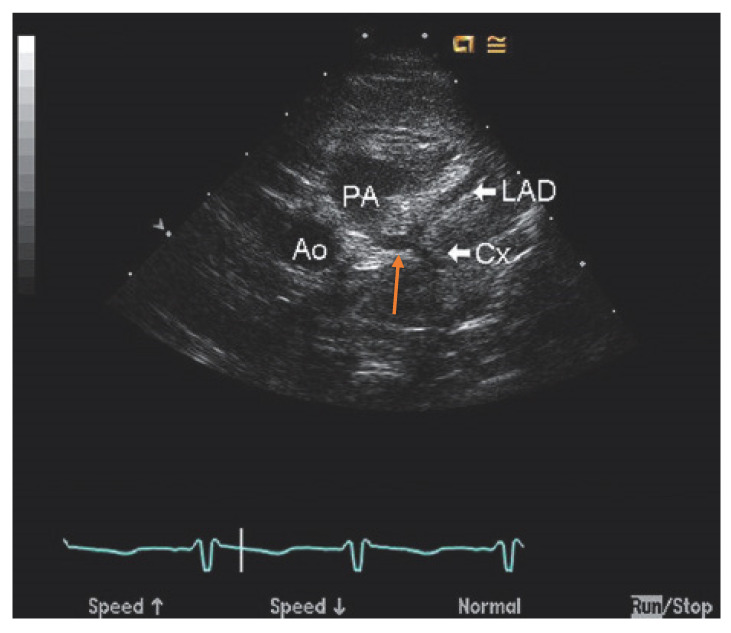
Echocardiography of anomalous left coronary artery from the pulmonary artery (ALCAPA) obtained from the parasternal short axis view at the base of the heart reveals anomalous origin of the left coronary artery (orange arrow) from the posterior aspect of the pulmonary artery (PA) far from the aorta (Ao). It divides into the left anterior descending artery (LAD) and left circumflex artery (Cx). Used with permission from Eidem, B. W.; Cetta, F.; O’Leary, P. W. *Echocardiography in Pediatric and Adult Congenital Heart Disease*, 3rd ed.; Wolters Kluwer: Alphen am Rhein, The Netherlands, 2021; pp. 514–530. [13].

**Figure 3 jpm-13-01561-f003:**
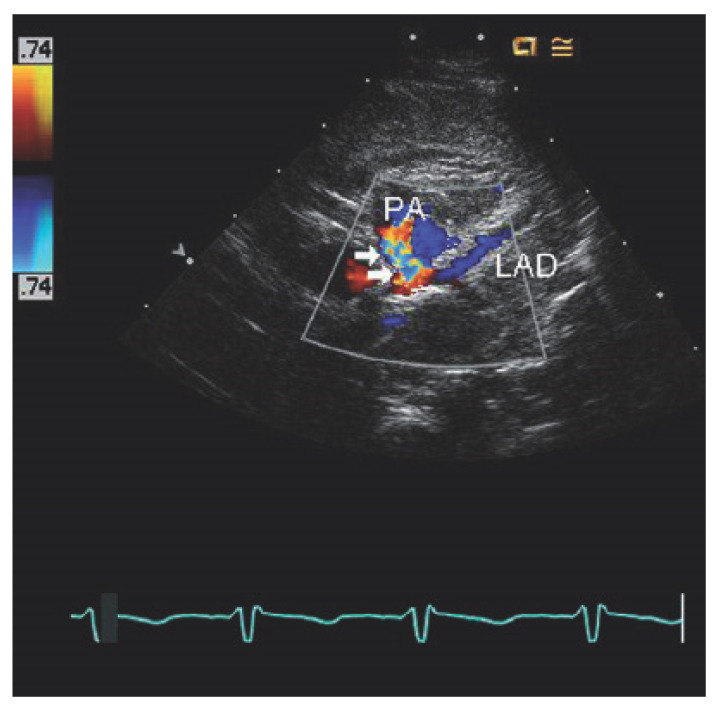
Echocardiography of anomalous left coronary artery from the pulmonary artery show-ing retrograde flow in the left anterior descending (LAD) coronary artery. Flow in the LAD is blue as it moves away from the transducer toward the pulmonary artery (PA), which is abnormal be-cause it should flow away from the aortic root (red Doppler signal) rather than toward it. A turbu-lent flow signal (arrows) is also seen in the pulmonary artery as the anomalous left coronary ar-tery empties into the low-pressure pulmonary artery. Used with permission from Eidem, Benjamin W., Jonathan Johnson, Leo Lopez and Frank Cetta. *Echocardiography in Pediatric and Adult Congenital Heart Disease*. Available from: Wolters Kluwer, (3rd Edition), 2021: 514-530. [13].

**Figure 6 jpm-13-01561-f006:**
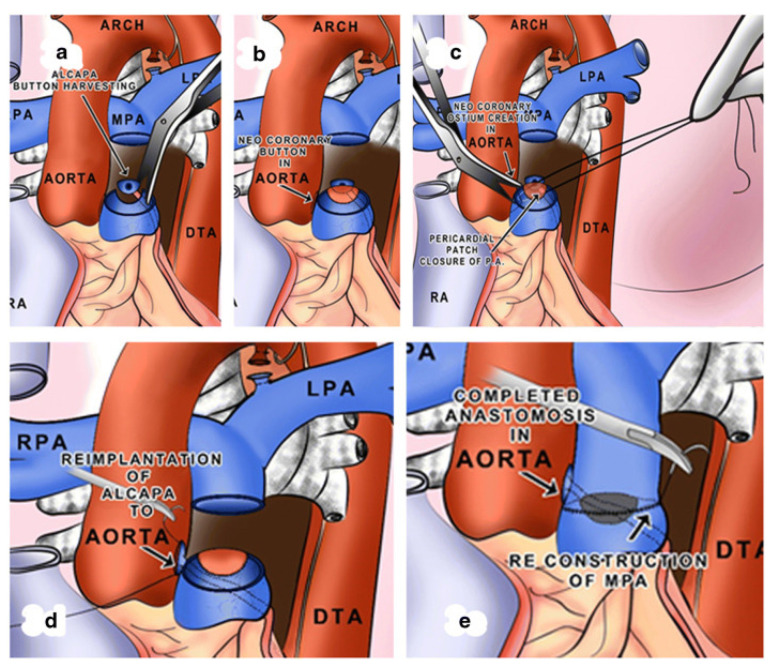
Schematic diagram outlining the coronary button transfer technique. (**a**) The main pulmonary artery (MPA) is transected with harvesting of the coronary button. (**b**,**c**) The MPA is reconstructed with a pericardial patch at the site of coronary button excision, and an aortotomy is created in the posterolateral aortic wall. (**d**) The left coronary artery is reimplanted end-to-side to the posterolateral aorta. (**e**) Re-anastomosis of the MPA. Used with permission from Mishra, A. Surgical management of anomalous origin of coronary artery from pulmonary artery. *Indian J. Thorac. Cardiovasc. Surg.* **2021**, *37*, 131–143. [19].

**Figure 7 jpm-13-01561-f007:**
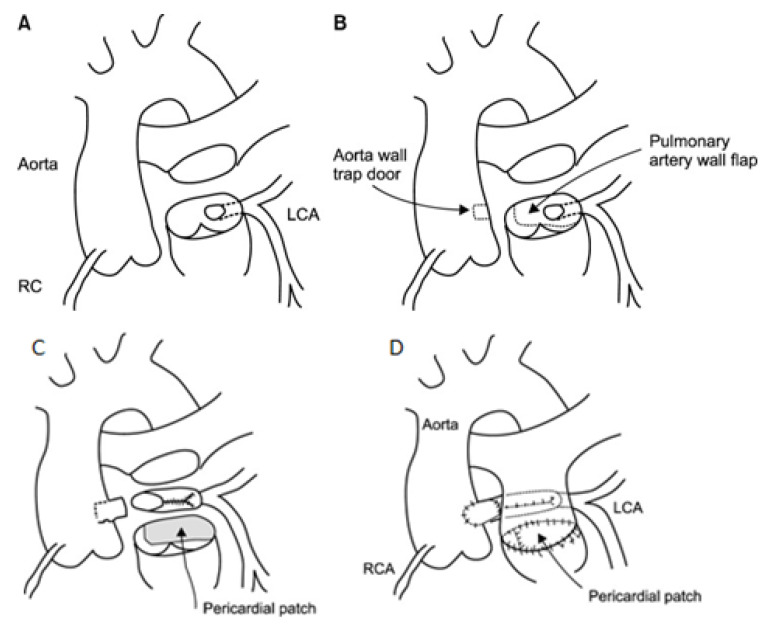
(**A**) The main pulmonary artery is transected above the origin of the left coronary artery (LCA). (**B**) The pulmonary artery flap and aortic trap door are designed as shown. (**C**) The LCA is anastomosed to the aortic trap door anteriorly and directly to the aorta posteriorly. (**D**) The pulmonary artery is repaired with autologous pericardium and the main pulmonary artery is reconstructed. LCA: left coronary artery; RC and RCA: right coronary artery. Used with permission from Kim YS, Lee M, Cho YH, Yang JH, Jun TG. An alternative surgical technique for repair of anomalous origin of the left coronary artery from the pulmonary arteryKorean J Thorac Cardiovasc Surg. 2014 Jun; 47(3):220-4. [21].

**Figure 8 jpm-13-01561-f008:**
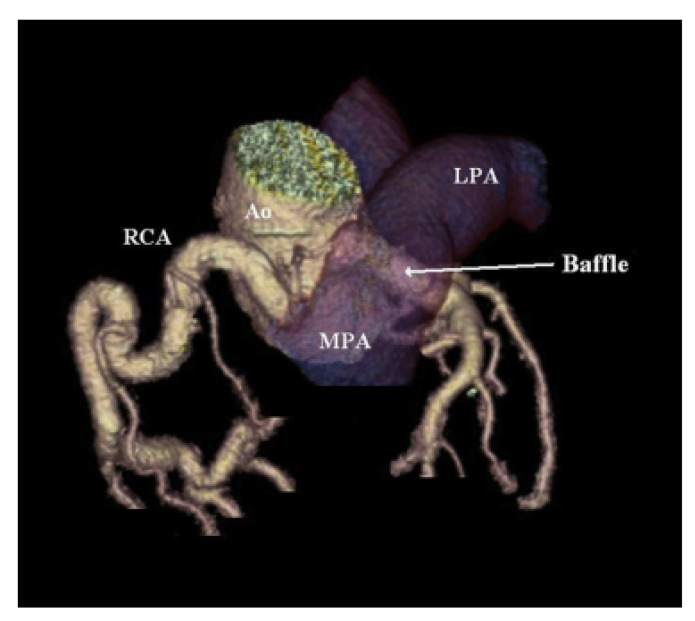
Three-dimensional-reconstructed image from coronary computed tomography angiogram demonstrating intrapulmonary baffle following Takeuchi repair method. Used with permission from Cramer, J.W.; Cinquegrani, M.; Cohen, S.B. Takeuchi repair of anomalous left coronary artery from the pulmonary artery. *J Cardiovasc. Comput. Tomogr*. **2015**, *9*, 457–458. [22].

**Table 1 jpm-13-01561-t001:** Comparison of different noninvasive imaging modalities in the diagnosis of anomalous left coronary artery from the pulmonary artery.

	Echocardiogram	Magnetic Resonance Imaging	Coronary Computed Tomography Angiography
Cost	Low	High	Medium
Availability	Readily available	Limited availability	Readily available
Scan Time	Quick	Long	Quick
Temporal Resolution	+++	++	+
Spatial Resolution	+	++	+++
Visualization of Nearby Structures	Limited	Good visualization	High visualization
Limitations	Operator-dependent and acoustic windows may be limited by patient’s body habitus	Sedation may be needed for patients with claustrophobia Dependent on patient following breathing commands	Ionizing radiation Iodinated nephrotoxic contrast Requires low heart rate
Features of Anomalous Left Coronary Artery from the Pulmonary Artery	Dilated and tortuous right coronary artery Lack of left coronary artery from the left aortic cusp Significant and dilated collateral vessels with arterial flow in the ventricular septum (lower Nyquist limit) Left ventricular systolic dysfunction with regional wall motion abnormalities Mitral valve regurgitation Enhanced echogenicity of the papillary muscles	Direct visualization of the left coronary artery from the main pulmonary artery Retrograde flow from the left coronary artery into the pulmonary artery (direction and flow can be quantified) Left ventricular dysfunction, regional wall motion abnormalities and left ventricular hypertrophy Mitral regurgitation Delayed subendocardial enhancement	Direct visualization of the left coronary artery from the main pulmonary artery Dilated right coronary artery with extensive collateral vessels (along epicardial surface or within the interventricular septum) Dilated bronchial vessels

+ refers to low; ++ medium; +++ high.

## Data Availability

Data is unavailable.

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
