# Peer review of "Anomalous Left Coronary Artery from the Pulmonary Artery: How to Diagnose and Treat"

_jpm, 2023, doi:10.3390/jpm13111561_

Round 1
Reviewer 1 Report
Comments and Suggestions for Authors
Blickenstaff et al.’s review on ALCAPA is well written and general in it’s scope. It is not clear who the target audience is and what is the aim or focus of the review. As it stands, it sits well as a book chapter in a general cardiology textbook. It should be stated what is the review methodology. A cursory search on Google Scholar brings up 243 publications on the subject in 2023. Most factoids are therefore recycled. It is also worrying that out of the 8 figures used, only one is original (Figure 5 – could have more of different orientations). Hence, lack of schematic consistency (Figures 6 vs 7). Perhaps it could be rewritten into 2 papers: 1. Diagnosis of ALCAPA, and 2. The management of ALCAPA, and review the literature accordingly with more original figures. The embryology of ALCAPA is also fascinating and should be illustrated step and step for clarity.
Author Response
Dear reviewer,
Thank you for the your insightful feedback regarding our manuscript. The narrative review was written to cover the basic diagnosis and management of anomalous coronary artery from the pulmonary artery (ALCAPA) with target audience of primary care providers, medical trainees, and community cardiologists. Pubmed was the database we queried and going back to 2010 there were only 50 reviews listed with most of them with limited scopes such as case reports/series, imaging findings, surgical intervention, late complications of repair, etc… Our manuscript is a general review focused only on ALCAPA across all ages and encompassing diagnostic testing, hallmark findings on imaging, treatment/surgical interventions and associated complications. Our manuscript does contain multiple figures which feature classic findings. Figures 2 and 3 are from one of our co-authors, Dr. Cetta. We do have images from our institution but we opted showcase (and paid to get copyright permission) ones that, in our opinion, were representative of the findings discussed in the manuscript (i.e., ECG findings, cardiac catheterization, etc….). We could have requested the surgical images to be redrawn at our institution but the timeline for their completion would be beyond the deadline set by the journal for publication of the special congenital heart disease issue. We did include additional details of the embryology as recommended-- keeping in mind the target the audience.
Thank you again
Reviewer 2 Report
Comments and Suggestions for Authors
Clear and interesting review of this rare coronary anomaly. Only little comment, the conclusion paragraph is very extensive, I suggest to transfer most of the text to the discussion. Conclusion is better like short and consise statement.
Author Response
Dear reviewer,
Thank you for your positive feedback. We had shortened the conclusion as you had requested. Attached is the modified manuscript with tracked changes.
Thank you again